# The layered costs and benefits of translational redundancy

**Parth K Raval[1]\*, Wing Yui Ngan[2], Jenna Gallie[2], Deepa Agashe[1]\***

[1]National Centre for Biological Sciences (NCBS-TIFR), Bengaluru, India; [2]Max Plank Institute for Evolutionary Biology, Plön, Germany

**Abstract** The rate and accuracy of translation hinges upon multiple components – including transfer RNA (tRNA) pools, tRNA modifying enzymes, and rRNA molecules – many of which are redundant in terms of gene copy number or function. It has been hypothesized that the redundancy evolves under selection, driven by its impacts on growth rate. However, we lack empirical measurements of the fitness costs and benefits of redundancy, and we have poor a understanding of how this redundancy is organized across components. We manipulated redundancy in multiple translation components of *Escherichia coli* by deleting 28 tRNA genes, 3 tRNA modifying systems, and 4 rRNA operons in various combinations. We find that redundancy in tRNA pools is beneficial when nutrients are plentiful and costly under nutrient limitation. This nutrient-dependent cost of redundant tRNA genes stems from upper limits to translation capacity and growth rate, and therefore varies as a function of the maximum growth rate attainable in a given nutrient niche. The loss of redundancy in rRNA genes and tRNA modifying enzymes had similar nutrient-dependent fitness consequences. Importantly, these effects are also contingent upon interactions across translation components, indicating a layered hierarchy from copy number of tRNA and rRNA genes to their expression and downstream processing. Overall, our results indicate both positive and negative selection on redundancy in translation components, depending on a species' evolutionary history with feasts and famines.

**\*For correspondence:**
parth_science@yahoo.com (PKR);
dagashe@ncbs.res.in (DA)

**Competing interest:** The authors declare that no competing interests exist.

## Editor's evaluation

The authors investigate the cost and benefits of maintaining seemingly redundant multiple copies of the translation machinery components. The authors demonstrate that while redundant multiple copies are beneficial in a nutrient-rich environment, maintaining these extra copies is costly and deleterious in a nutrient-poor environment. This explains why copy numbers of translation machinery genes are under selection according to the environmental niche an organism occupies. The work is very important and the findings exciting and supported by compelling evidence, in particular, the fitness gain upon deletion of translation genes in poor conditions is an insightful observation.

## Introduction

In the early 1960s, the degeneracy of the genetic code was revealed in the context of multiple synonymous codons encoding a given amino acid. A large body of work since then has uncovered an astonishing degree of redundancy in the translation apparatus. The redundancy is often qualitative, whereby some components can functionally compensate for others. This includes the pool of tRNA molecules that read mRNA codons and deliver the appropriate amino acid during translation. tRNAs with different anticodons may read distinct synonymous codons but carry the same amino acid (tRNA isoacceptors) and are thus functionally degenerate. In addition, tRNA modifying enzymes (MEs) post-transcriptionally alter specific tRNAs ('target' tRNAs, i.e. targets of modification), allowing them to

**eLife digest** Translation is the process by which cellular machines called ribosomes use the information encoded in genes to make proteins . Every organism requires two types of RNA molecules to make new proteins: ribosomal RNAs (rRNAs, which form part of the ribosome) and transfer RNAs (tRNAs, which transport the amino acid molecules that form proteins to the ribosomes). These RNA molecules are coded in the genome, but different organisms have different 'copy numbers': some genomes contain just a few copies of each of these genes, while others have thousands.

This apparent redundancy – the presence of several copies of the same gene – is puzzling because it is costly to make and maintain DNA and RNA. This leads to an important question: how does redundancy in these important genes (coding for tRNAs and rRNAs) evolve? The answer is key to understanding how one of the most fundamental cellular processes, the making of proteins from DNA, has evolved.

A possible reason for organisms to have many copies of the genes required to make proteins is to allow rapid translation, which allows cells to divide faster, and populations of cells to grow more quickly. However, this would likely mean that, when nutrients are scarce, carrying and translating many copies of the same gene would become a burden on the cell. Raval et al. set out to test this idea by measuring the costs and benefits of seemingly redundant translation components.

To do this, Raval et al. deleted some of the redundant gene copies in the bacterium *Escherichia coli* and asked if that changed bacterial growth. The experiments showed that when nutrients were plentiful, cells with more copies of the genes (high redundancy) were better able to use the nutrients and divide rapidly. However, when nutrients were limited, bacteria with extra gene copies divided more slowly, showing that the extra genes are indeed a big burden on the cell.

Raval et al. propose that nutrients available in the environment ultimately determine whether redundancy of the translation machinery is a blessing or a curse. This suggests that the redundancy and underlying growth strategies of different organisms are forged by their experiences of feast and famine during their evolutionary past.

Importantly, by testing the joint effect of many different molecules involved in translation, Raval et al. uncovered several strategies that may maximize bacterial growth and protein production. Their results could thus be useful for optimizing the synthesis of important products that use growing cells as factories – from beer to insulin – where the rate of growth is critical.

read codons that are otherwise decoded by 'non-target' tRNAs that are not modified by MEs (*Grosjean, 2009*). Hence, some non-target tRNAs could be redundant because their function can be carried out by target tRNAs after they get modified. For instance, in *Escherichia coli* the codon CCG can be decoded by the non-target tRNA$_{CGG}$ (encoded by the gene *proK*), but also by the target tRNA$_{UGG}$ (encoded by *proM*) after the U at position 34 is modified to cmo$^5$U by the cmo modification pathway (in principle rendering the gene *proK* redundant). MEs are thus critical for maximizing cellular decoding capacity (*Diwan and Agashe, 2018*; *Grosjean et al., 2010*), and maintaining translational capability requires either a very diverse tRNA pool or the presence of MEs that allows for a compact tRNA gene set (*Diwan and Agashe, 2018*; *Marck and Grosjean, 2002*; *Rocha, 2004*; *Wald and Margalit, 2014*). In addition, quantitative redundancy is conferred by high gene copy number (GCN), such that multiple genes can perform identical functions. For instance, bacterial genomes often carry many copies of genes encoding tRNAs with the same anticodon (tRNA isotypes) (*Chan and Lowe, 2009*). Similarly, cells typically have many copies of rRNA genes (*Roller et al., 2016*), which govern ribosome availability (*Nomura et al., 1980*). Thus, the translation machinery is predicted to be functionally redundant at many levels, with qualitative (multiple codons, tRNA isoacceptors, and MEs) as well as quantitative (GCN) redundancy. However, in many cases, the predicted functional redundancy is not experimentally verified, and it remains unclear whether it influences fitness. We asked: what are the fitness costs and benefits of translational redundancy, and under what conditions do they manifest? Could these costs and benefits explain the evolution of a highly redundant translation apparatus?

In bacteria, selection for rapid growth (often facilitated by nutrient availability) is thought to be an important force that shapes the evolution of many translation components. The maximum possible growth rate is determined by translation efficiency, which in turn depends on the concentrations

of ribosomes and tRNAs (*Ehrenberg and Kurland, 1984*; *Hu et al., 2020*; *Kurland and Ehrenberg, 1987*; *Kurland and Hughes, 1996*). Across species, there are striking positive correlations between maximal growth rate and the number of tRNA and rRNA genes (*Dethlefsen and Schmidt, 2007*; *Ikemura, 1985*; *Rocha, 2004*; *Roller et al., 2016*; *Vieira-Silva and Rocha, 2010*; *Weissman et al., 2021*). Thus, we expect that GCN redundancy in key translation components should be especially beneficial during rapid growth in a nutrient-rich niche. In contrast, under nutrient limitation, expressing redundant genes should be costly because translational output remains constrained by nutrients. However, this overarching growth rate-dependent selection may shape the redundancy of translation components differentially. For instance, selection should have the maximum impact on rRNA, whose GCN shows the strongest correlation with growth rate (*Rocha, 2004*; *Roller et al., 2016*). Ribosomal RNAs constitute up to 85% of all RNA in rapidly growing *E. coli* (*Bremer and Dennis, 1996*) and their concentration is most limiting for translation, likely because rRNA constitutes the core catalyst of ribosome assembly (*Bremer and Dennis, 1996*). During fast growth, cellular rRNA abundance increases by ~250%, whereas total tRNA increases only by ~80% (*Dong et al., 1996*). Consequently, the cellular ratio of tRNAs to rRNAs decreases under rapid growth, indicating greater investment in rRNA (*Dittmar et al., 2004*). Indeed, deleting rRNA operons in *E. coli* reduces fitness in rich media, but improves fitness in poor media (*Condon et al., 1995*; *Gyorfy et al., 2015*; *Stevenson and Schmidt, 2004*).

In contrast, the growth rate-dependent impacts of redundancy in tRNA GCN remain largely unexplored, with the exception of initiator tRNA genes in *E. coli*. As predicted, in this case, the loss of some gene copies is deleterious in rich media and advantageous in poor media (*Samhita et al., 2014*). However, the growth impact of elongator tRNA GCN is not known. The paucity of data for tRNA redundancy is glaring because bacteria show enormous variation in tRNA pools, driven by evolutionary changes in tRNA GCN as well as MEs (*Ayan et al., 2020*; *Bedhomme et al., 2019*; *Diwan and Agashe, 2018*; *Saks et al., 1998*; *Wald and Margalit, 2014*). Furthermore, the impacts of redundancy are predicted to vary substantially across different tRNA genes. For instance, mature tRNA pools are biased towards 'major' tRNAs that connect frequently used amino acids and codons, and are more strongly correlated with growth rate (*Berg and Kurland, 1997*; *Dong et al., 1996*) and rRNA GCN (*Mahajan and Agashe, 2018*) compared to minor tRNAs. The loss of redundancy in these tRNAs should impose a larger fitness cost, a hypothesis that remains untested. Thus, despite strong comparative evidence for growth rate-driven selection, our understanding of the evolution and impacts of redundancy in translation components is far from complete.

While the fitness costs and benefits of redundancy should ultimately be shaped by nutrient availability, redundancy may itself arise via different mechanisms for different translation components, and hence selection may shape the components in distinct ways. For instance, with rRNAs, the loss of GCN redundancy is buffered by strong compensatory upregulation of 'backup' gene copies. An *E. coli* strain with deletion of 6 out of 7 rRNA operons can produce about half of the normal levels of rRNA (*Asai et al., 1999*). As a result, the deletion of a few rRNA genes only moderately reduces the growth rate in rich media (*Gyorfy et al., 2015*), though the direction of the effect is reversed in poor media (see above). Are tRNA pools similarly regulated? While this is possible (e.g. in response to nutrient availability *Fessler et al., 2020*; *Sørensen et al., 2018*), we have no data on compensatory regulation of the remaining tRNA copies after gene loss. Furthermore, tRNA redundancy is modulated not only by GCN but also by MEs that allow target tRNAs to perform the function of non-target tRNAs. Thus, tRNA gene loss could be buffered by the regulation of other tRNA gene copies, and/or by the action of MEs. Interestingly, fast-growing bacteria tend to have low tRNA diversity (*Rocha, 2004*), with their decoding capacity likely maintained by the action of multiple tRNA modification pathways (*Diwan and Agashe, 2018*). Hence, the fitness consequences of tRNA gene loss should be contingent on the availability of ME backups. Conversely, the joint deletion of non-target tRNAs and MEs is predicted to be more costly than the loss of either component alone (*Diwan and Agashe, 2018*; *Wald and Margalit, 2014*). Thus, the mechanisms that mediate redundancy as well as the interactions between translation components are important to fully understand the evolution of translational redundancy.

The patterns noted above suggest a hierarchical organization, whereby redundancy in some components and genes is more important than others (e.g. rRNAs vs. tRNAs, and major vs. minor tRNA isotypes). However, as discussed above we have very limited empirical evidence for the fitness consequences of redundancy in different translation components, particularly in the case of tRNA

**Figure 1.** Summary of experimental manipulation of redundancy in translational components. Target transfer RNAs (tRNAs): tRNA isotypes that are targets of modifying enzymes (MEs) (i.e. are post-transcriptionally modified by a tRNA modifying enzyme), expanding the repertoire of codons they can read. Non-target tRNAs: tRNA isotypes that are not acted upon by MEs. The codons recognized by these tRNAs can also be read by modified target tRNAs, rendering the non-target tRNAs redundant. Symbols represent qualitative differences across sets rather than the exact number or diversity of redundant components. Further details of strains in each set are given in *Supplementary file 1A*.

pools. We addressed these gaps by analyzing the nutrient-dependent impact of changing redundancy in multiple translational components, alone as well as in combination. Specifically, we tested the following predictions: (1) redundancy in tRNA and rRNA GCN, on the whole, should be important to maintain rapid growth, and the benefits of increased redundancy should be proportional to the achievable growth rate (2) broadly, a reduction in rRNA GCN should have stronger fitness impacts than tRNA GCN (3) across tRNAs, the loss of redundancy should be most impactful for major tRNA isotypes, and for non-target tRNAs when combined with the loss of a relevant ME (4) the fitness impact of reduced redundancy should increase with the severity of the loss, e.g., due to the deletion of multiple gene copies or multiple translational components. We worked with *E. coli* because it has a highly redundant translation machinery (*Diwan and Agashe, 2018*; *Wald and Margalit, 2014*) that allowed us to test the impacts of successive losses of redundancy at the level of rRNA genes, tRNA pools, and tRNA modifying enzymes. We first show that, as expected, many components of the translation machinery are indeed redundant with respect to fitness. We then test our predictions by measuring the context-dependent costs and benefits of this redundancy. Our results reveal layered factors that may have shaped the evolution of the translation machinery in bacteria.

## Results

### Altering redundancy in translation components

Prior work demonstrates the functional redundancy of some bacterial translation components with respect to translation rate or accuracy. However, it remains unknown whether and under what conditions this functional redundancy has fitness consequences. The genome of *E. coli* MG1655 (wild type, WT) encodes 42 tRNA isotypes with varying copy numbers (total 86 tRNA genes) (*Chan and Lowe, 2016*), five tRNA modification pathways (*Diwan and Agashe, 2018*) that modify the 34th base of the

tRNA or first base of the anticodon, and seven rRNA genes in distinct operons (*Quan et al., 2015*). We reduced redundancy in translation components in three ways (*Figure 1*, *Supplementary file 1A*). (1) We generated 23 distinct mutant strains of WT that represented a total of 28 deleted tRNA genes, with 20 strains carrying single tRNA deletions and three strains carrying multiple tRNA deletions. These strains denoted a direct genomic loss of redundancy, potentially altering the cellular tRNA pool (sets I, II and III, *Figure 1*, *Supplementary file 1A*). (2) Post-transcriptional modification enhances wobble pairing by adding anticodon loop modifications to target tRNAs. Hence, non-target tRNAs are made redundant by modified tRNAs. To reduce this form of redundancy, we deleted key enzymes (MEs) within four tRNA modification pathways of WT (set IV, *Figure 1*, *Supplementary file 1A*), as well as some of their non-target tRNAs (set V, *Figure 1*, *Supplementary file 1A*) and a target tRNA in one case. (3) Finally, to lower redundancy in rRNA genes, we used strains carrying 1–4 rRNA operon deletions, including deletions of interspersed tRNA genes (*Quan et al., 2015*) (set VI, *Figure 1*, *Supplementary file 1A*). In one of the strains with four rRNA operons deletions, we made additional tRNA deletions to determine the fitness impact when the levels of both rRNA and tRNA are reduced (set VII, *Figure 1*, *Supplementary file 1A*). Overall, we used 43 mutant strains covering 15 amino acids, 33 tRNA genes, three tRNA modifying systems, and four rRNA operons (*Figure 1*, *Supplementary file 1A*).

## The loss of tRNA redundancy has highly variable growth impacts

Of our strains, 15 represented the deletion of single tRNA genes. All genes except one (*proL*) were predicted to be redundant because the genome has other gene copies encoding the same tRNA isotype (set I in *Figure 1*, *Supplementary file 1A*). In some cases, the original GCN was small so that our manipulation left a single redundant copy (*phe U/V* and *ser W/X*). In five other strains, we deleted single tRNA genes that appear redundant because MEs should allow other (target) tRNAs to perform their function (set II, *Supplementary file 1A*). Given the expected functional backups for deletions in sets I and II, we predicted that the loss of redundancy should have relatively weak fitness consequences. Indeed, in complex rich media (LB and TB), the growth rate of these strains was largely similar to the WT, with the highest impact representing ~15% change in growth rate ($R_{rel}$ values between 0.85 and 1.15; $R_{rel}$ is the ratio of mutant growth rate to WT growth rate, so $R_{rel} >1$ indicates faster growth of mutant). Only 10 of 20 strains showed a significant difference in at least one of these complex rich media, six with faster growth and four with slower growth than WT (sets I and II in *Figure 2A*, *Figure 2—figure supplement 1A*, *Figure 2—figure supplement 2A*, *Supplementary file 2*). Interestingly, deleting the only genomic copy of *proL* had negligible effects on growth (*Figure 2A*), possibly because the relevant codon is used very rarely (*Supplementary file 1A*). On the other hand, deleting different genes encoding the same tRNA isotype (e.g. the four genes for tRNA-$_{GUU}$-Asn) did not have identical growth impacts, corroborating previous work suggesting that copies of the same tRNA contribute differently to translation (*Dittmar et al., 2004*). These differences could potentially arise due to genomic location – prior work shows that tRNA gene expression is correlated with distance from the origin of replication (*Ardell and Kirsebom, 2005*; *Hu and Lercher, 2021*). However, we did not find a correlation between the fitness impacts of gene deletion and distance from the origin of replication, across copies of the same tRNA isotype or across all tRNA genes (*Figure 2—figure supplement 3*).

As predicted, a more severe reduction in redundancy via deletion of multiple tRNA gene copies (leaving only one backup copy of many, set III, *Figure 1*) reduced the growth rate substantially, with a mean $R_{rel}$ of ~0.75 (i.e., ~25% change) (*Figure 2A*, *Figure 2—figure supplement 1A*, *Figure 2—figure supplement 2A*). Over half of the 20 strains also showed a significant difference in the length of the lag phase, though the direction of the effect varied across strains and media (*Figure 2—figure supplement 2B*, *Figure 2—figure supplement 4A*). Consistent with the growth rate results, set III strains showed the maximum increase in lag phase length. However, barring a few exceptions, the loss of tRNA redundancy had a negligible impact on growth yield regardless of the severity of the manipulation (*Figure 2—figure supplement 2C*, *Figure 2—figure supplement 5A*).

Next, we tested the fitness of tRNA deletion strains in more permissive rich media with easy-to-use sources of carbon (glucose) and amino acids (casamino acids) ('GA'), where the WT growth rate is similar to that in complex rich media. Here, tRNA gene loss was uniformly deleterious, with 18 of 20 strains showing significantly slower growth (GA 1.6, sets I and II, mean $R_{rel}$ = 0.84 and 0.88

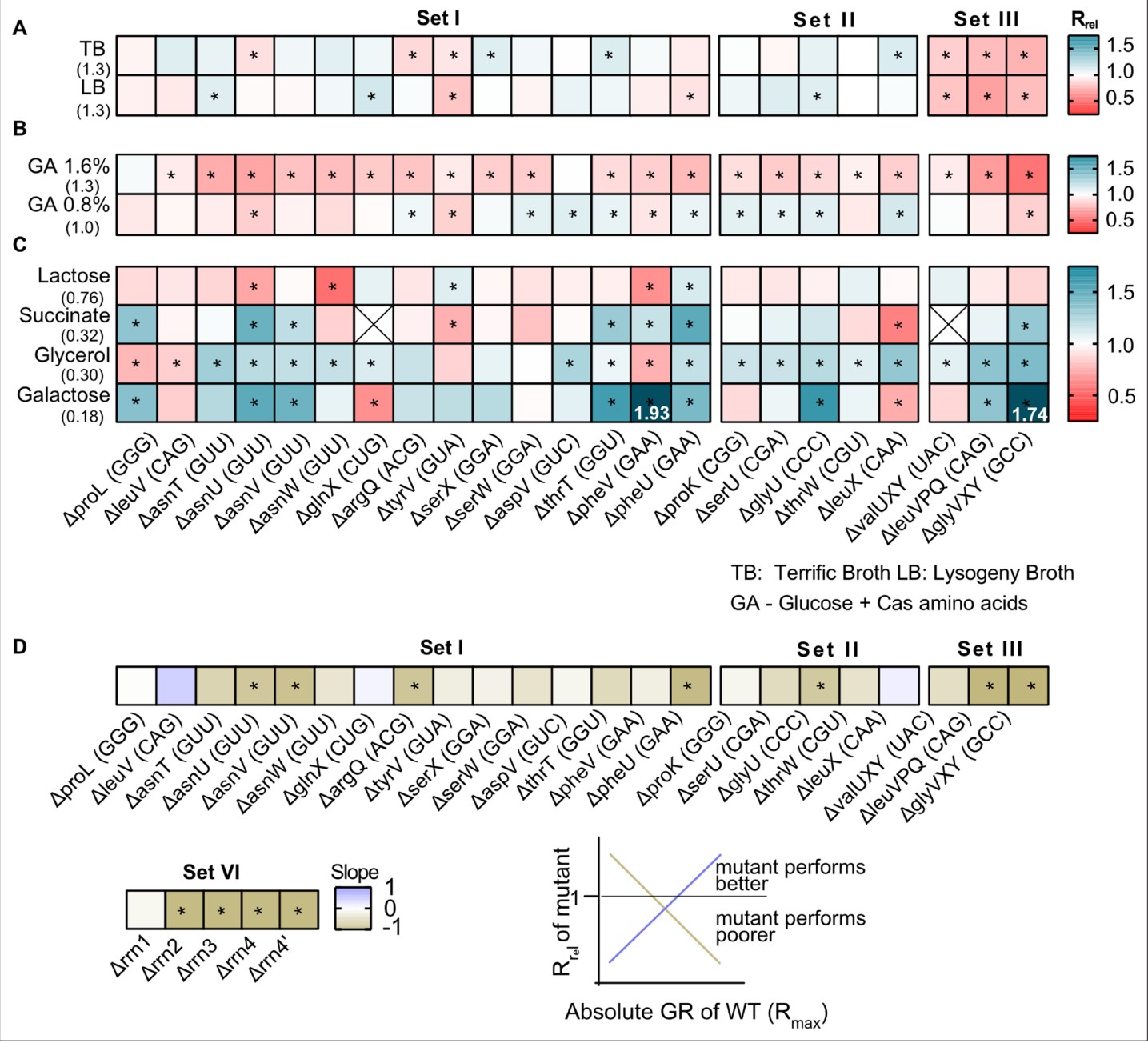

**Figure 2.** The fitness impact of loss of transfer RNA (tRNA) redundancy varies with nutrient availability. (**A–C**) Heat maps show the relative growth rate of tRNA gene deletion strains ('mutant') ($R_{rel}$ = $GR_{mutant}/GR_{WT}$) in different growth media (see *Figure 2—source data 1* file for raw data and *Supplementary file 2* for statistics). The anticodon encoded within each deleted tRNA gene is indicated in parentheses on the x-axis. The absolute exponential growth rate (doublings/hour) of WT in each medium is indicated in parentheses on the y-axis. Box colors indicate the impact of each gene deletion (red: $R_{rel}$ <1, mutant grows more slowly than WT; blue: $R_{rel}$ >1, mutant grows faster than WT; n=3–4 biological replicates per strain per medium); statistically significant differences from WT are indicated by asterisks (ANOVA with Dunnet's correction for multiple comparisons). Boxes with an 'X' indicate cases where strains failed to grow exponentially. Panels show $R_{rel}$ in (**A**) complex rich media (**B**) permissive rich media (M9 salts supplemented with indicated concentrations of glucose and cas amino acids) (**C**) poor minimal media (M9 salts) with the indicated carbon source but no cas amino acids. In two cases, we observed very high $R_{rel}$ values in galactose and these are indicated numerically. (**D**) For each mutant we estimated Spearman's rank correlation between the growth rate impact of the gene deletion ($R_{rel}$) and the respective maximal wild type (WT) growth rate ($R_{max}$) across eight growth media (*Figure 2—figure supplement 5*). The heat map shows the slope of this correlation for each mutant. Darker colors indicate a stronger negative relationship, i.e., a higher cost of redundancy in poor media. Statistically significant non-zero slope values are indicated by asterisks. Data for mutants with rRNA operon deletions (across six growth media) are also included in this panel (see *Figure 6B*).

The online version of this article includes the following source data and figure supplement(s) for figure 2:

*Figure 2 continued on next page*

*Figure 2 continued*

**Source data 1.** Data associated with *Figure 2*.

**Figure supplement 1.** Growth curves ($OD_{600}$ vs. time) for representative strains in rich and poor media.

**Figure supplement 2.** Mean relative growth rate ($R_{rel}$), length of lag phase ($L_{rel}$), and yield ($K_{rel}$) of mutant strains across all tested media.

**Figure supplement 3.** Correlation between distance from oriC and fitness effect of transfer RNA (tRNA) gene deletion.

**Figure supplement 4.** Impact of transfer RNA (tRNA) deletion on length of the lag phase.

**Figure supplement 5.** Fitness impact of transfer RNA (tRNA) deletions on growth yield.

**Figure supplement 6.** Rescue of select transfer RNA (tRNA) deletion mutants by gene complementation from a plasmid.

**Figure supplement 7.** Correlation between absolute growth rate of the wild type (WT) and relative growth rate of mutants.

respectively; *Figure 2B*, *Figure 2—figure supplement 1B*, *Figure 2—figure supplement 2A*, *Supplementary file 2*). Reducing the glucose and casamino acid concentration reversed this effect (GA 0.8, sets I and II, mean $R_{rel}$ = 0.98 and 1.02 respectively; *Figure 2B*), suggesting that nutrient availability determined both the direction and uniformity of the impact of tRNA redundancy. As with complex rich media, strains with a severe loss of redundancy tended to show the largest reduction in fitness (set III, *Figure 2B*, *Figure 2—figure supplement 1B*, *Figure 2—figure supplement 2A*). However, the impacts on other growth parameters were more variable. In GA 1.6, 6 of 20 strains (sets I and II) showed a significantly shorter lag phase than WT while five showed a longer lag phase; but in GA 0.8, only one strain had a longer lag phase and 12 strains entered the exponential growth phase faster than WT (*Figure 2—figure supplement 4B*). Concomitantly, we observed little change in the growth yield of these strains, with only 3–4 strains showing a significant difference in either medium (*Figure 2—figure supplement 5B*). Unlike the patterns in complex rich media, set III strains did not show stronger effects on either lag phase length or growth yield (*Figure 2—figure supplement 2B–C*). Overall, in media where easily accessible nutrients are plentiful, even a small loss of tRNA redundancy strongly hindered rapid growth but had weak and/or inconsistent effects on the lag phase and growth yield.

Finally, we measured fitness in poor media where nutrients should severely limit translation, and maintaining tRNA redundancy may be costly. When using lactose (which reduces WT growth rate to ~50% of LB), the loss of redundancy had a weak and variable impact, with only 5 of 23 strains (across sets I–III) showing a significant difference from WT (*Figure 2C*, *Supplementary file 2*). However, in poorer media containing succinate, glycerol, or galactose (where WT growth rate is reduced to ~10–25% of LB), tRNA gene deletions were often beneficial (7, 15, and 9 out of 23 strains, respectively) and only 2–3 strains had slower growth than the WT (sets I–III, *Figure 2C*, *Figure 2—figure supplement 2A*). Again, set III strains tended to show the maximum benefit of tRNA loss (*Figure 2C*, *Figure 2—figure supplement 2A*). Although, the impacts on other growth parameters varied across strains and growth media, most strains had a shorter lag phase (20 of 23 strains) and a higher yield (14 of 23 strains) in at least one poor medium (*Figure 2—figure supplement 2B–C*, *Figure 2—figure supplement 4C*, *Figure 2—figure supplement 5C*). Hence, the loss of redundancy appeared to be generally beneficial in poor media.

Finally, we found that the low growth rate of two of the tRNA deletion strains (ΔglyU and ΔthrW) was rescued by plasmid-borne tRNA copies, confirming that the observed fitness effects were indeed explained by tRNA loss (*Figure 2—figure supplement 7*). Overall, these results indicated that a severe reduction in tRNA redundancy amplified the fitness impacts of tRNA deletion, but the magnitude and direction of the effects varied substantially across growth media. However, in all growth media, the fitness impacts were generally similar for sets I and II (*Figure 2—figure supplement 2A–C*), indicating that the nature of the backup available to maintain tRNA pools (redundant gene copies vs. ME activity) does not alter the impact of tRNA gene loss.

## tRNA redundancy is beneficial during rapid growth but costly under nutrient limitation

The results above showed that the fitness impacts of tRNA gene loss depend qualitatively on the growth medium. To test whether these patterns are quantitatively explained by growth limits set by nutrient availability, for each engineered strain we estimated the relationship between the relative impact of a loss of redundancy ($R_{rel}$) and the maximum attainable WT growth rate ($R_{max}$), across all

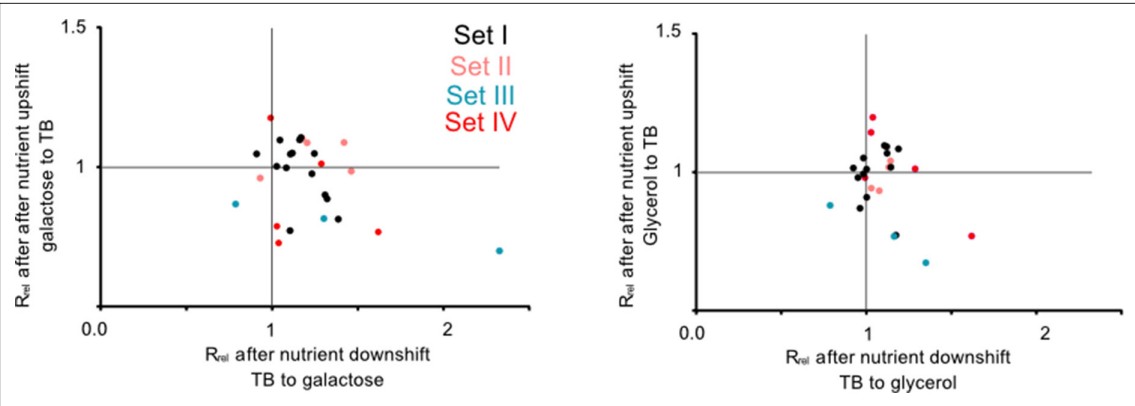

**Figure 3.** Redundant transfer RNA (tRNA) gene copies are beneficial during nutrient fluctuations. Relative growth rate of mutants (sets I to IV, **Figure 1**) upon a nutrient downshift ($R_{rel}$ on the x-axis) and an upshift ($R_{rel}$ on the y-axis). $R_{rel}$ was calculated relative to wild type (WT) (growing in identical conditions), as described in **Figure 2** (see **Figure 3—source data 1** file for raw data and **Supplementary file 2** for statistics). Strains were transferred between a rich medium (TB, terrific broth) and a poor medium (M9 salts +0.6% glycerol or 0.05% galactose). Each data point represents the mean $R_{rel}$ for a mutant (n=4 biological replicates), with the color indicating the set to which it belongs (**Figure 1**). Mutants in the bottom right quadrant perform better under a nutrient downshift (i.e. when moving from a rich to poor medium), but more poorly than WT under a nutrient upshift (i.e. moving from a poor to rich medium). Mutants in the top right quadrant perform better than WT in both types of nutrient fluctuations. Statistics are reported in **Supplementary file 2**.

The online version of this article includes the following source data for figure 3:

**Source data 1.** Data associated with **Figure 3**.

growth media tested (data from **Figure 2A–C**). Since WT has the highest level of redundancy and presumably the weakest internal limits on translation rate, we expected that the WT $R_{max}$ reflects the nutrient capacity of each growth medium (i.e. externally placed limits on growth). Of 28 strains (including five rRNA operon deletions described later in the results section), 26 showed a negative correlation between $R_{max}$ and $R_{rel}$, with a significant relationship in 11 cases (**Figure 2D**, **Figure 2—figure supplement 6**). Thus, tRNA and rRNA gene loss tends to be more beneficial (i.e. redundancy is more costly) under conditions of low nutrient availability, when the maximum possible growth rate is constrained.

This pattern was further supported by experiments performed during nutrient shifts, where cultures in the exponential growth phase were transferred from rich to poor media and vice versa. Note that this setup differs from the previous growth measurements (**Figure 2**) where late stationary phase cultures grown overnight were transferred to either rich or poor media. Of the 27 tRNA and rRNA deletion strains tested, all but one had a growth rate that was comparable to WT (16 strains) or higher than WT (10 strains) after transitioning from rich to poor media (i.e. during a nutrient downshift, note data distribution along the x-axis in **Figure 3**; **Supplementary file 2**). In contrast, after a nutrient upshift, 11 strains showed significantly slower growth in one or both pairs of media, and only two showed a significantly faster growth than WT (note data distribution along the y-axis in **Figure 3**; **Supplementary file 2**). Thus, gene loss is beneficial during a nutrient downshift but deleterious in a nutrient upshift. These patterns were also most consistent for strains in sets III and VI (described later in the results section), corroborating our results from constant environments where we observed large impacts of redundancy for the same strains. Strains in the bottom right quadrant of **Figure 3** are especially interesting because they represent cases where the loss of redundancy is beneficial in a nutrient downshift but deleterious in a nutrient upshift. Hence, these genes should be important when ramping up translation in a nutrient-rich environment. In this category, we observed one strain each from sets I and II, and 4 strains from set VI (**Supplementary file 2**). Thus, redundancy in tRNA genes can be beneficial during rapid growth but is generally costly in poor media where nutrients are limited.

## Expression from redundant gene copies cannot compensate for the loss of tRNA genes

Recall that when tRNA redundancy was lowered to an extreme (set III, **Figure 1**), cells retained at least one copy of each tRNA gene and could potentially compensate for tRNA gene loss by upregulating

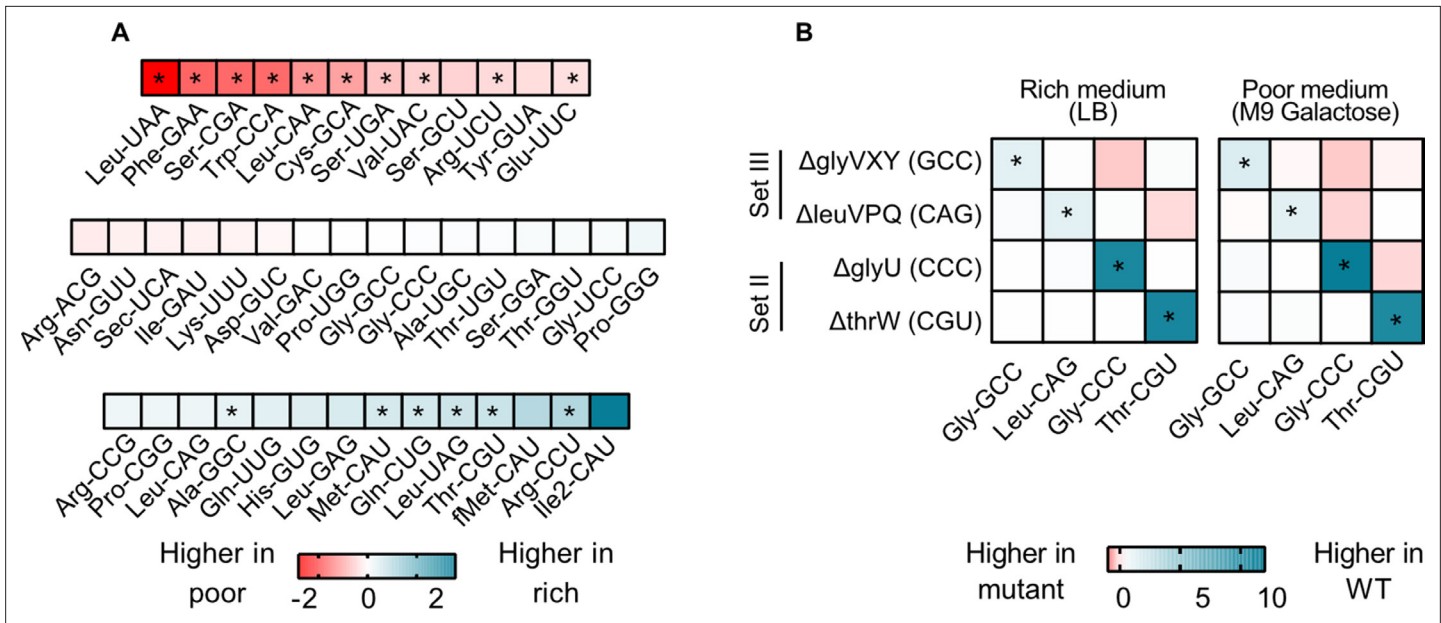

**Figure 4.** transfer RNA (tRNA) gene copy number changes confer stronger control over the proportional contribution of tRNA isotypes to the mature tRNA pool, than does tRNA gene regulation. The proportional contribution of all 42 tRNA isotypes in the mature tRNA pool was measured for wild type (WT) and four tRNA deletion mutants in a poor medium (M9 +0.05% galactose) and a rich medium (LB) (n=3 biological replicates per medium per strain) using YAMAT-seq (see *Figure 4—source data 1* file for raw data and *Supplementary file 2* for statistics). The change in tRNA proportions across strains or across media was then calculated as mean log₂ fold change with respect to the 'reference' as indicated in each comparison. Significant differences are shown with asterisks (pairwise Wald tests with Benjamini-Hochberg correction for multiple comparisons; *Supplementary file 2*). (**A**) The relative proportion of all WT tRNA isotypes in poor medium, compared to rich medium. Isotype is indicated on the x-axis. Blue indicates a higher proportional contribution in rich medium, and red indicates a higher contribution in poor medium. (**B**) The impact of tRNA gene deletion on the proportions of focal isotype tRNAs (indicated on the x-axis), in WT vs. mutant strains (tRNA gene deletions are indicated on the y-axis), in a rich and a poor medium. Darker blue colors indicate lower proportional contributions in the mutant, and darker red indicates higher proportional contributions compared to WT.

The online version of this article includes the following source data and figure supplement(s) for figure 4:

**Source data 1.** Data associated with *Figure 4*.

**Figure supplement 1.** Proportional contribution of transfer RNA (tRNA) isotypes to the total tRNA pool in wild type (WT) *E. coli* in different media.

**Figure supplement 2.** Impact of transfer RNA (tRNA) deletion on the composition of the tRNA pools in rich and poor media.

this backup copy (or copies). However, these deletion strains paid a substantial fitness cost in rich media (*Figure 2A–B*, *Figure 2—figure supplement 1A*, *Figure 2—figure supplement 2A*), suggesting that such upregulation (if present) could not fully compensate for the reduction in tRNA GCN. Conversely, the large fitness benefit of deleting the same tRNA genes in poor media (*Figure 2C*) suggests that these genes are not sufficiently downregulated in poor media, with cells paying a maintenance cost.

To test these hypotheses, we focused on four strains from sets II and III where we observed strong fitness effects. We measured the relative contribution of each tRNA isotype to the mature tRNA pool in WT and the tRNA deletion strains, in a rich (LB) and a poor medium (M9 galactose). In the WT, 26 of 42 (~62%) tRNA isotypes did not show a significant difference in relative levels across media (*Figure 4A*, *Figure 4—figure supplement 1*). This is consistent with minimal in-built regulation of tRNA expression, as suggested by prior work showing a remarkably consistent composition of the tRNA pool across growth rates (*Ardell and Kirsebom, 2005*). Notably, of the tRNA isotypes whose loss imposed a large fitness cost in rich media (set III, as well as a few from set I; *Figure 2A*), none showed a higher relative abundance in rich media compared to poor media (*Figure 4A*; *Supplementary file 2*). Thus, although the absolute cellular amounts of all tRNAs in *E. coli* increase in rich media (*Dong et al., 1996*), media-dependent changes in the composition of the WT tRNA pool do not reflect the fitness consequences of deleting tRNA isotypes.

Next, we compared the composition of tRNA pools between WT and tRNA deletion strains. We expected that tRNA gene deletion would reduce the proportional contribution of the respective tRNA isotype to the total pool unless the deletion was compensated by expression from the remaining

gene copies. In all four gene deletion mutants, the contribution of the focal (deleted) tRNA isotype to the total tRNA pool was lower than in the WT tRNA pool, in both rich and poor medium (*Figure 4B*, *Figure 4—figure supplement 2*). Thus, expression from backup gene copies was not sufficient to compensate for the loss of the deleted tRNA genes. Together, these results suggest that change in gene copy number allows a much stronger (in this case, also more beneficial) response to the nutritional environment, compared to what can be achieved by differential gene expression (including processes such as gene regulation or tRNA degradation).

Interestingly, in the tRNA gene deletion strains, in addition to the focal isotype, the fractional contribution of many other tRNA species also differed from the WT (*Figure 4B*). Since YAMAT-seq measures relative tRNA isotype levels, when we deleted genes whose products are typically highly represented in the mature tRNA pool (e.g. ΔglyVXY and ΔleuVPQ) we expected to automatically alter the relative contribution of other tRNA isotypes. However, the difference between WT and tRNA gene deletion strains varied systematically across media (*Figure 4—figure supplement 2*) and was also evident with the deletion of tRNA genes whose products show intermediate levels of contribution to the mature tRNA pool. Hence, these differences likely reflected transcriptional regulation or downstream processing. Broadly, the loss of tRNA gene copies led to stronger changes in tRNA pool composition in the rich medium, even though it did not restore fitness completely. This further indicated that in rich media, gene copy number

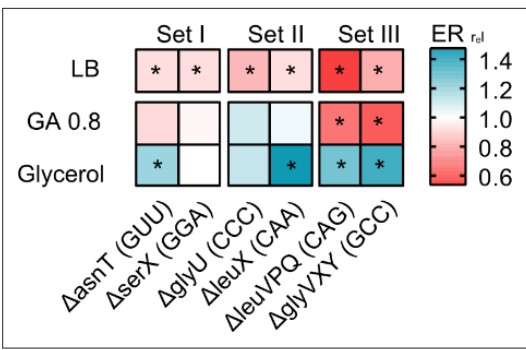

**Figure 5.** Loss of redundant transfer RNA (tRNA) genes decreases translation output during rapid growth, but increases translational output in poor media. The heat map shows the translation capacity of tRNA gene deletion mutants in rich (LB), permissive (M9 +0.8% GA) and poor medium (M9 +0.6% glycerol), measured as the relative protein elongation rate (ER, increase in the length of ß-galactosidase protein per unit time after induction; $ER_{rel} = ER_{mutant}/ER_{WT}$; n=2 biological replicates per strain per medium, see *Figure 5— source data 1* file for raw data and *Supplementary file 2* for statistics). Red indicates lower reporter protein production in the mutant per unit cell density (i.e. reduced translation capacity), and blue indicates increased translation capacity relative to wild type (WT). Significant differences are indicated with asterisks (ANOVA with Dunnet's correction for multiple comparisons).

The online version of this article includes the following source data and figure supplement(s) for figure 5:

**Source data 1.** Data associated with *Figure 5*.

**Figure supplement 1.** Impact of transfer RNA (tRNA) deletions on translation capacity.

is limiting, whereas in poor media nutrients are a major limiting factor. Note that as expected, deletion of the single-copy, non-target tRNA genes *thrW* or *glyU* caused near-elimination of these tRNA isotypes from the tRNA pool (*Figure 4—figure supplement 2*). However, the fitness impact of these deletions was weak (*Figure 2*), likely due to the action of MEs that allowed other (i.e. target) tRNAs to compensate for the loss of the deleted single-copy tRNAs. Thus, MEs indeed serve as a backup and render non-target tRNAs redundant. Overall, these data strongly suggest that moderate to severe loss of redundancy at the gene copy level is not fully rescued by expression from backup gene copies.

## Redundant tRNAs do not contribute to translation when nutrients are limited

We showed above that the WT over-produces many tRNA isotypes in poor media, potentially explaining its low fitness relative to the tRNA gene deletion strains. We suspected that although in rich media such 'surplus' tRNAs contribute to translation, this may not happen in poor media where growth is limited by nutrient availability rather than translation efficiency, and levels of charged amino acids drop (*Dittmar et al., 2005*; *Elf et al., 2003*). Thus, in rich media, the loss of tRNA genes should decrease translation; but in poor media, this effect should be weak. We therefore estimated translational output in a subset of our strains, by measuring the translation elongation rate of the native beta-galactosidase protein during the log phase of growth. As predicted, in a rich medium (LB) all strains with low redundancy had a significantly slower elongation rate than WT (*Figure 5*). In a permissive

medium (GA), elongation rates were usually not significantly different from WT, a pattern that was also observed for another reporter protein (GFP, *Figure 5—figure supplement 1*). However, in a poor medium (M9 glycerol), elongation rates were often higher than WT (*Figure 5*), indicating that the loss of tRNA genes had a net beneficial impact on translation elongation. Again, the effect of tRNA gene deletion on translation rate increased with the magnitude of the loss of redundancy, with set III strains showing the largest effect size (*Figure 5*, *Figure 5—figure supplement 1*). Overall – as expected from the correlation between growth rate and translation rate – these results mirror the impacts of tRNA redundancy on fitness (*Figure 2*). Thus, under nutrient limitation, redundant tRNAs are expressed (with cells paying the cost of expression), but these tRNAs do not contribute to growth because they do not increase translational output enough to compensate for the cost of expression.

## Loss of redundancy in multiple translation components reveals layered fitness impacts

Recall that tRNA gene deletion strains with severely reduced redundancy (set III, with only 1 gene copy remaining) showed stronger fitness effects than strains in set I and II. Furthermore, strains in set II had similar fitness as those in set I, potentially because the fitness impact of the tRNA deletion (which also reduced the levels of the deleted tRNA isotype) was masked by the presence of MEs. Thus, the fitness impact of redundancy generally increases with the magnitude of the loss, as predicted by comparative evidence across genomes; but this has not been explicitly demonstrated. To do so, we further lowered translational redundancy by simultaneously deleting multiple translation components.

Loss of the modifying enzyme-coding genes *cmoA*, *cmoB*, *mnmG*, and *tgt* (set IV, *Figure 1*) significantly reduced growth rate in all media (*Figure 6A*, *Supplementary file 2*). Consistent with our prediction, the combined fitness effect of ME deletion and non-target tRNA gene deletions (set V, *Figure 1*) was stronger than the effect of deleting only the non-target tRNA genes, with all five tested co-deletions showing a significantly higher effect in at least one medium (*Figure 6A*). However, the impact of co-deletions was statistically indistinguishable from the effect of deleting MEs alone, in all except the leucine triple deletion in LB and Glycerol (*Supplementary file 2*). Conversely, and as expected, co-deletion of the ME *tgt* and one copy of its target tRNA gene (*asnU,* which showed a strong phenotype) had a significantly different impact from the deletion of the target tRNA alone in all media. This suggested that the effect of co-deletion is largely driven by ME loss, provided that the co-deleted tRNA is not a target tRNA. As observed with tRNA gene deletions alone (*Figure 2*), in rich media the loss of ME redundancy tended to be deleterious, whereas in poor media the effects were more variable and included cases where gene loss was beneficial. Note that the joint importance of redundancy in ME and non-target tRNAs is observed in both rich and poor media, with more instances of beneficial effects in the latter. Thus, these results confirmed that MEs serve as important backups when the diversity of the tRNA pool is depleted and that a reduction in redundancy (via tRNA GCN and/or ME loss) is generally beneficial in poor media.

Next, we tested the combined effect of altering redundancy in rRNA and tRNA genes. As noted earlier, deleting rRNA operons simultaneously removes some tRNA genes located in the operon, so that even a single rRNA operon deletion is effectively a co-deletion. However, all the tRNA genes deleted in this manner had multiple genomic backup copies (*Supplementary file 1A*). Given the strong regulatory compensation of rRNA loss observed in prior work (*Asai et al., 1999*; *Elf et al., 2003*; *Quan et al., 2015*), it was not surprising that the deletion of up to three rRNA operons (along with up to four tRNA genes) had a very weak impact on growth rate in rich media. However, in poor media, the fitness impact was evident even with the loss of only two rRNA operons (and three tRNA genes). Mimicking the patterns observed for specific tRNA gene deletions, a more severe loss of rRNA and tRNA redundancy (from one to four rRNA operon deletions, set VI, *Figure 1*, *Supplementary file 2*) was detrimental in rich media (set VI, *Figure 6B*) and increasingly beneficial in poor media (set VI, *Figure 6B*).

Further loss of redundancy – via simultaneous deletion of four rRNA operons (including seven tRNA genes) and extra-operonic tRNA genes (set VII, *Figure 1*, *Supplementary file 1A*) – had mixed effects on growth rate. In every case, the impact of the rrn4 deletion was significantly greater than the tRNA deletion alone (*Supplementary file 2*), but the effect of tRNA co-deletions varied across media. In rich media, only 1 of the 6 cases tested showed a significant additional fitness reduction upon co-deleting rRNA and extra-operonic tRNA. However, in poor media, tRNA loss led to an additional

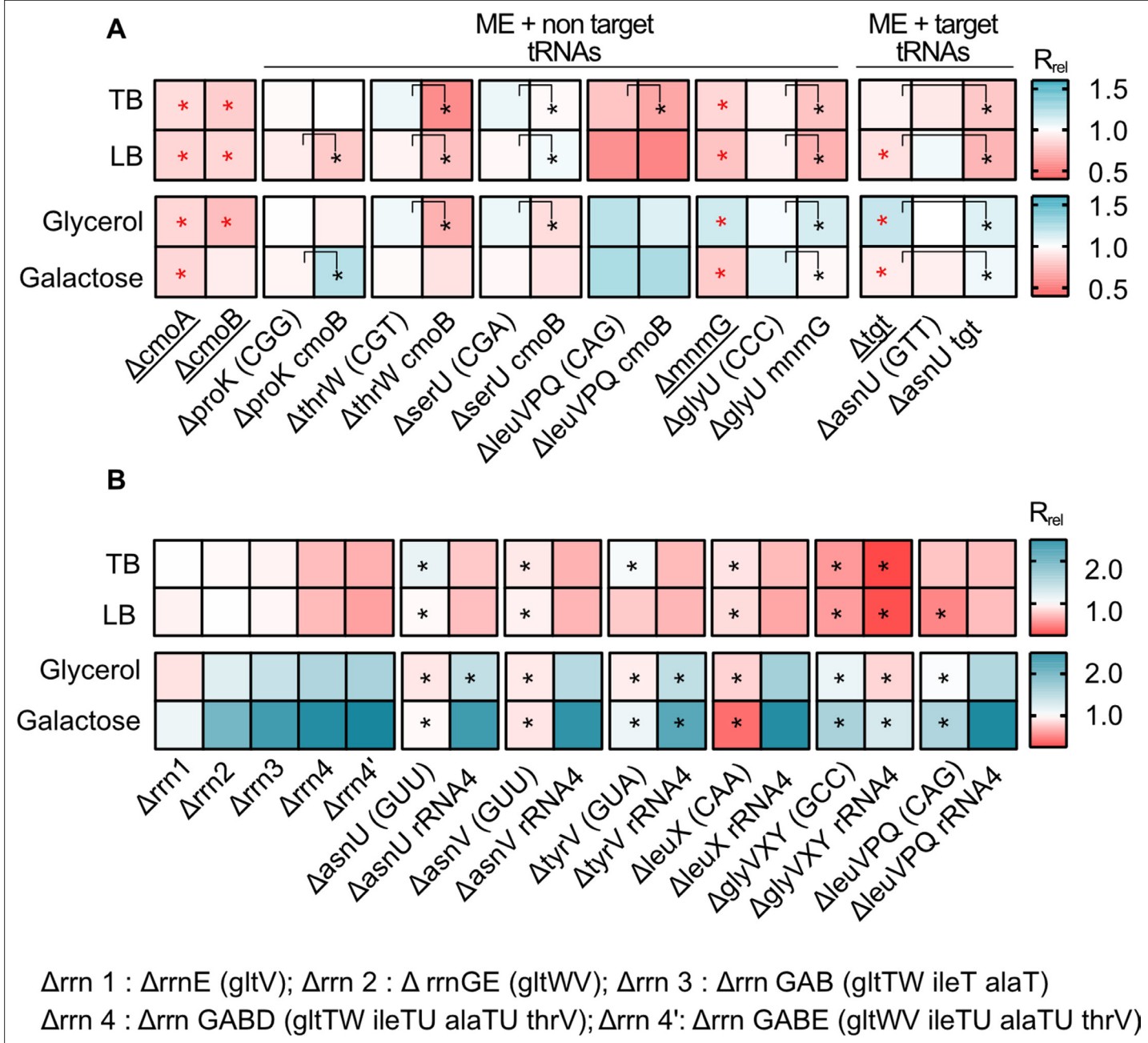

**Figure 6.** Impacts of manipulating redundancy in multiple translation components are highly variable. Impact of transfer RNA (tRNA), rRNA and modifying enzyme (ME) gene deletion on growth rate in different media, as described in *Figure 2*. Box colors indicate the impact of gene deletion on growth rate relative to wild type (WT) (red: $R_{rel}$ <1, mutant grows more slowly than WT; blue: $R_{rel}$ >1, mutant grows faster than WT; n=3–4 biological replicates per strain per medium, see *Figure 6—source data 1* file for raw data and *Supplementary file 2* for statistics). Panels show $R_{rel}$ for (**A**) co-deletion of MEs and tRNA genes. The anticodon of each deleted tRNA gene is indicated in parentheses on the x-axis. MEs were either deleted alone (underlined strains) or co-deleted with the respective non-target and target tRNAs. ME deletions strains were compared with the WT to establish statistically significant differences, indicated with a red asterisk. For co-deletions of MEs and non-target tRNAs, comparisons are shown between non-target tRNA deletion and ME +non target tRNA codeletion (as indicated by the asterisks and vertical square brackets). For co-deletions of MEs and target tRNA, comparisons are shown between ME deletions and ME +target tRNA co-deletion (as indicated by the asterisks and vertical square brackets). Other statistical comparisons, not shown in the figures, are in *Supplementary file 2*. (**B**) Co-deletion of rRNAs and tRNAs. rRNA operons are abbreviated as shown in the key; tRNA genes that were deleted as part of the operon are indicated in parentheses. Additional tRNAs (outside the rRNA operon) that were deleted in combination with rrn4 are indicated on the x-axis. Asterisks indicate statistically significant impacts ΔrRNA4 vs. ΔrRNA4+tRNA deletion, (ANOVA with Dunnet's correction for multiple comparisons, *Supplementary file 2*).

The online version of this article includes the following source data for figure 6:

**Source data 1.** Data associated with *Figure 6*.

fitness benefit in 3 out of 6 co-deletions. Thus, when growth is strongly limited by the availability of rRNA (and hence mature ribosomes), the lack of tRNA is less detrimental to translation and growth rate. Conversely, when growth is limited by both nutrients and rRNAs, shedding tRNAs appears to be additionally beneficial.

Together, our results lead to the following conclusions. First, we find that the simultaneous deletion of multiple copies of tRNA genes or rRNA genes has more severe fitness consequences than the loss of single gene copies. Second, increasing the severity of reduction in translational redundancy via co-deletion of MEs and tRNA genes amplifies the fitness consequences of losing redundancy. Lastly, when nutrients are in plenty, rRNA becomes limiting and the loss of tRNA genes has little additional impact on fitness; but when nutrients are limiting, shedding different translation components (rRNA and tRNA) additively increases fitness.

## Discussion

The process of protein synthesis is central to life and is especially important to understand bacterial evolution given the direct link between translation, growth rate, and fitness. Translation rate is affected by several genomic components (e.g. tRNAs, rRNAs, tRNA modifying and charging enzymes) and environmental factors (e.g. nutrient availability). Comparative analyses show that the genomic components have different degrees of redundancy across taxa, and suggest that this redundancy (in the fitness context) should be shaped by the strength of ecological selection for rapid growth (*Rocha, 2004*; *Roller et al., 2016*). While lateral gene transfer and gene deletion/duplication make translation components labile even in the absence of selection, the correlation between maximum growth rate and redundancy of tRNA or rRNA GCN suggests a strong role for selection (*Rocha, 2004*; *Roller et al., 2016*), likely imposed by nutrients from the environment. Hence, nutrients ultimately limit translation. Cells can meet this environmental limit and maximize fitness by shaping genomic factors to achieve the maximum attainable translational output. Such modulation of the cellular machinery should occur at physiological (short-term) as well as evolutionary timescales. For instance, cells can

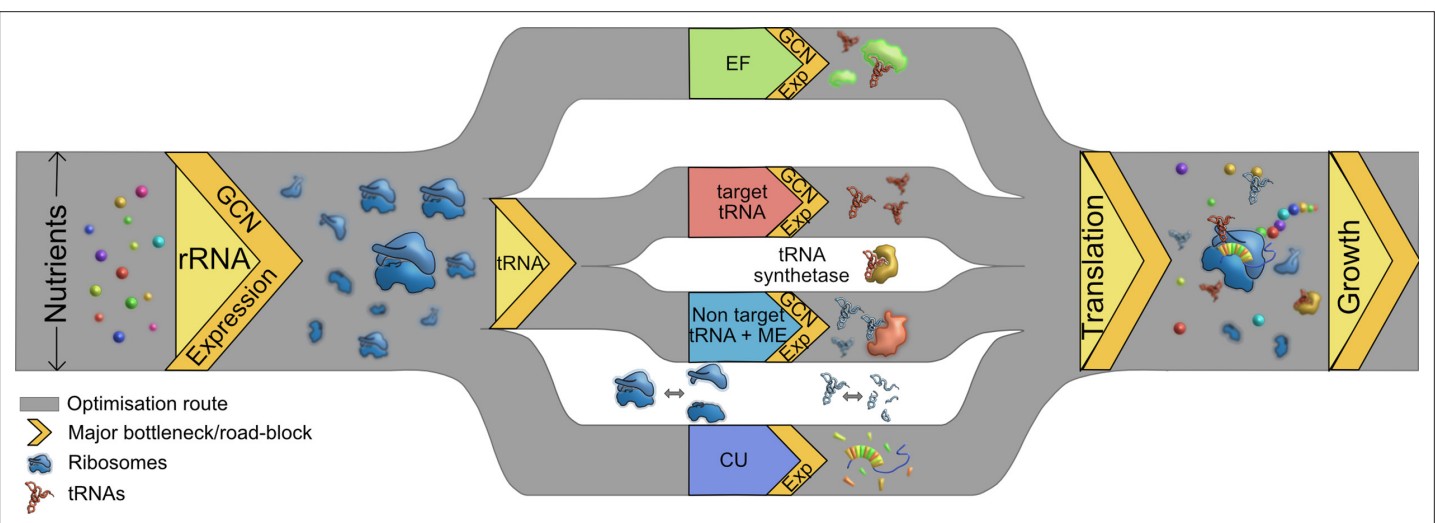

**Figure 7.** Multiple routes to translational adaptation. Proposed highway model for the layered, hierarchical organisation of translation components showing multiple potential routes for optimizing translational output, based on the current study and previous work. The width of the routes is constrained primarily by available nutrients, which sets an upper limit to translational output; and the redundancy of various components evolves under this constraint. Within the confines of what is achievable under given nutrient conditions, different components may adapt in different order, and to different degrees. Important components that serve as bottlenecks are shown in yellow, from left to right in the proposed order of importance. rRNA should adapt first (via gene expression change in the short term and gene copy number, GCN, over long term), contributing to ribosome synthesis. Next, transfer RNA (tRNAs) may adapt via two possible parallel routes: by modulating the copy number or expression of target tRNAs, or non-target tRNA +MEs (see Introduction). Elongation factors (EF) can adapt by similar routes. Simultaneously, a stronger match between the tRNA pools and codon usage (CU) of highly expressed genes will enhance translational output. Components that are not directly tested or discussed extensively here but are likely to adapt via the general principles derived here are shown in the white spaces: tRNA synthetases and ribosomal and tRNA disassembly/ degradation. Together, all these factors determine translation output, growth, and fitness.

control translation rates via rapid regulation of translation components (*Wilusz, 2015*), including via degradation of ribosomes and tRNAs during nutrient starvation (*Fessler et al., 2020*; *Sørensen et al., 2018*). Across-species patterns of rRNA and tRNA GCN are consistent with such selection, as discussed in the Introduction. Similar arguments can also be made for redundancy across other translation components. Together, this suggests that the environment sets the limits of translation, according to which natural selection shapes the genomic redundancy of translational components. However, empirical evidence for a common underlying selection pressure shaping redundancy across various translational components has been missing.

Here, we provide such evidence, showing that broadly speaking, several components of the *E. coli* translation apparatus are indeed functionally redundant and that the costs and benefits of this redundancy vary with nutrient availability. When nutrients permit rapid growth, the loss of redundancy in both tRNAs and rRNAs is detrimental, as these components become limiting for translation. This is especially true for multiple deletions of abundantly used components such as rRNAs, MEs that modify many different tRNAs, and frequently used tRNAs (such as glycine and leucine). These results support prior predictions of larger fitness consequences following the loss of major tRNAs that read abundant codons and respond strongly to fast growth (*Dong et al., 1996*), or tRNAs that make larger contributions to the tRNA pool (*Bloom-Ackermann et al., 2014*; *Kanaya et al., 1999*). The observed variability in fitness impacts across tRNAs is therefore at least partially explained by the relative use of different codons. Our results also support the broad prediction that high tRNA levels should be most critical during rapid growth (*Mahajan and Agashe, 2018*; *Vieira-Silva and Rocha, 2010*), when the correlation between tRNA levels and gene copy number is strongest (*Dong et al., 1996*). Conversely, when nutrients are limiting, both rRNA as well as tRNA gene loss is beneficial. Importantly, we show that the expression cost of high translational redundancy of WT *E. coli* is not met by increased trans-lation rate in poor media, and imposes a substantial net fitness cost.

While these observations suggest that nutrient availability can guide the evolutionary optimization of the cost-to-benefit ratio of the bacterial translation machinery, interactions (and potential hierar-chies) amongst different components suggest multiple routes of optimization (*Figure 7*). We observe that the loss of rRNA genes is generally more impactful than the loss of tRNAs; and when rRNAs are limiting, the additional loss of tRNAs has a relatively weak effect. We suggest that this is because rRNAs set the first internal limit on translation rate, as predicted by prior work (see the Introduction). However, further experiments are necessary to separate the independent impacts of rRNA and tRNA genes linked within operons. Interestingly, even in complex bacterial communities, the addition of resources enriched for taxa with more rRNA copies (*Wu et al., 2017*) and the growth rate response to nutrient addition was positively correlated with rRNA copy number (*Li et al., 2019*). The next internal limiting factor for translation appears to be available tRNA pools, determined by the nested impacts of tRNA gene copy number, transcriptional regulation of gene expression, and MEs. Overall, tRNA gene copy number has a stronger impact on translation and fitness than the regulation of different isotype copies, corroborating prior results with yeast (*Percudani et al., 1997*). However, in the special case of non-target tRNAs, the relevant MEs have a stronger impact than the tRNA genes. This is not surprising given the predicted functional redundancy between MEs and non-target tRNAs (*Diwan and Agashe, 2018*; *Grosjean et al., 2010*). As a result of this layering, a severe loss of tRNA redun-dancy becomes important for fitness only when many copies of an isotype are lost, or when a non-target tRNA is co-deleted with a relevant modifying enzyme. Conversely, these results predict that an increase in redundancy (e.g. due to occupation of a more nutrient-rich niche and selection for rapid growth) may occur via increasing isotype gene copy numbers, increasing non-target tRNAs, or gaining a relevant modifying enzyme. These predictions from our *E. coli* data closely match the patterns observed in comparative analyses across bacteria: selection for rapid growth is strongly correlated with gene copy number (*Rocha, 2004*), and lineages may lose either MEs or non-target tRNAs, but not both (*Diwan and Agashe, 2018*).

Together with prior work, our results also suggest that fast-growing organisms such as *E. coli* have evolved to rely strongly on gene copy numbers to maximize translation rates. Presumably, such species are either able to bear the costs of maintenance of surplus tRNA genes during periods of slow growth, or across longer evolutionary timescales the extra tRNA copies provide a net benefit. We therefore suggest a model whereby bacterial growth rate is primarily limited by external nutrient availability, then by rRNA molecules, and finally by tRNA pools (determined by tRNA GCN and MEs,

and secondarily via tRNA gene regulation). Thus, the layered costs and benefits of high translational redundancy – both within and across distinct components – are ultimately determined by the environmental context. Importantly, this model predicts that prolonged selection for rapid growth should cause successive evolutionary changes in the redundancy of different translation components, with the general order of events determined (and parallel routes offered) by the hierarchical layering of components (*Figure 7*).

We hope that future work will test this model and enrich it by considering additional translational components and factors that may drive their evolution. For instance, genome GC content is strongly associated with tRNA GCN and diversity across the bacterial phylogeny (*Diwan and Agashe, 2018*; *Wald and Margalit, 2014*), as well as with codon bias (*Hershberg and Petrov, 2010*). An understanding of the ecological and evolutionary pressures that drive shifts in GC content would thus be useful to understand the impact of selection for rapid growth on genome GC. Parallel to our results, prior work shows that the two copies of *tuf* genes – encoding the *E. coli* translation elongation factor EF-Tu – also have distinct, medium-dependent impacts on growth rate (*Zuurmond et al., 1999*). It would thus be interesting to include EF-Tu redundancy in our model. It is also worth considering other evolutionary processes that can alter redundancy, such as genetic drift that may facilitate the loss of MEs (*Diwan and Agashe, 2018*). In addition, here we have focused on selection acting on translation rate, primarily tested using growth rate. However, our results hint at interesting effects of the loss of redundancy on other growth parameters such as yield and lag phase that may be orthogonal to growth rate. These measurements had a low resolution in our study, but explicit and better analysis of such impacts may reveal the effects of selection in niches where yield or survival (rather than growth rate) determine fitness. Finally, we note that selection may act on other cellular functions performed by some translational components (*Shepherd and Ibba, 2015*), or on translational accuracy (*Gingold and Pilpel, 2011*). While global mistranslation can provide fitness benefits in stressful contexts (*Jones et al., 2011*; *Samhita et al., 2020*), the nature and strength of selection acting on translational accuracy and the relationship between translation rate and accuracy remain poorly understood (*Drummond and Wilke, 2009*). Nonetheless, prior work suggests that both tRNA pools and MEs influence translation accuracy (*Manickam et al., 2016*) and protein aggregation (*Fedyunin et al., 2012*). Thus, selection for accuracy could also shape the evolution of tRNAs and MEs. We therefore suggest that expanding the layers of organization of translational components in our model will prove fruitful in gaining a deeper understanding of the evolution of translational redundancy.

In summary, our experiments demonstrate that several components of the translation machinery are redundant in *E. coli*, the costs and benefits of which vary based on nutrient availability – an environmental variable that likely shaped the redundancy in the first place. Our results support the broad idea that translational limits imposed by different components and their interactions generate multiple translational optima and make many paths feasible (*Grosjean et al., 2014*; *Higgs and Ran, 2008*), depending on the selective context (*Rocha, 2004*). We propose a model with hierarchies of translation components that may allow for the maximization of translational output and fitness. It remains to be seen to what extent these hierarchies operate across diverse taxa, and which of the possible parallel routes are taken during the course of bacterial evolution in nature. Further studies on the layers and hierarchies connecting translational components will thus shed light on the molecular toolkits underlying evolutionary transitions between slow versus rapid translation.

## Methods
### Generating strains
We made all gene deletions in *E. coli* MG1655, which we refer to as the wild type (WT). tRNA deletions were made using Red recombinase, slightly modifying the Datsenko-Wanner method (*Datsenko and Wanner, 2000*) with longer homology regions of 60–100 bases to increase the probability of recombination. In all but one case (ΔglyVXY) we removed the Kanamycin marker inserted during recombination. We confirmed all strains had marker-less deletions by PCR followed by Sanger sequencing (primers given in *Supplementary file 1B*) and Next Generation Sequencing (Illumina HiSeq PE150, >30 x coverage). We used P1 transduction to transfer ME deletions received from CGSC (Keio collection) to our WT strain, conducting additional rounds of transduction to make further tRNA deletions

as required. Similarly, we combined rRNA deletion strains (from CGSC) with tRNA deletions using P1 transduction. We stored glycerol stocks of each strain at –80 °C.

To combine ME and rRNA deletions with tRNA deletions, we used P1 transduction (**Thomason et al., 2007**). We grew recipient strains overnight and resuspended them in an equal volume of MC buffer (100 mM MgSO$_4$.7H$_2$O and 5 mM Cacl$_2$.2H$_2$O) and added 50 µl of phage lysate from the donor strain. After incubating at 37 °C (20 min), we washed the cells twice with 0.1 M citrate buffer (0.06 M citric acid C$_6$H$_8$O$_7$ and 0.04 M sodium citrate dihydrate C$_6$H$_9$Na$_3$O$_9$). After the second wash, we resuspended the cells in 1 mL LB with 20 mM sodium citrate and incubated at 37 °C (60–90 min). Finally, we pelleted the cells, resuspended in 100 µl of citrate buffer and 5 mM sodium citrate, plated them on LB agar with Kanamycin (30 µg/ml), and incubated overnight at 37 °C.

We removed the kanamycin cassette that replaced the deleted gene using plasmid pCP20 (**Datsenko and Wanner, 2000**). Briefly, we grew transformants (mutant +pCP20) at 30 °C, 18 h in LB medium (with ampicillin 100 µg/ml) and streaked them on LB agar plate with ampicillin. We patched individual colonies on LB agar, LB +Amp, and LB +Kan and incubated overnight at 37 °C. We confirmed the loss of the Kan marker (for all except ΔglyVXY) by PCR-amplifying and Sanger sequencing those colonies that had lost antibiotic resistance (indicating loss of the Kan cassette and the Amp-marked plasmid), and grew only on LB agar.

For all experiments, we used independently grown colonies or cultures to serve as biological replicates. Hence, unless mentioned otherwise, reported sample sizes refer to biological replicates.

## Measuring growth parameters

We inoculated strains in LB (Lysogeny Broth, Difco) from individual colonies grown from freezer stocks, and incubated cultures at 37 °C with shaking at 180 rpm for 14–16 hr (preculture). For growth rate measurement, we sub-cultured 1% v/v in 48 well microplates (Corning) in the appropriate growth medium: LB, TB (Terrific Broth, Sigma) or M9 minimal medium (M9 salts, 1 mM CaCl2, 2.5 mM MgSO4) supplemented with specific carbon and nitrogen sources ('GA': glucose and cas amino acids, either 1.6% w/v or 0.8% w/v each as specified in the results and figures; or carbon sources alone: lactose 0.05% w/v, galactose 0.05%, pyruvate 0.3% w/v, succinate 0.3% w/v, or glycerol 0.6% w/v). We measured growth rate (*r*) as the change in optical density (OD) read at 600 nm every 20 min or 45 min (for rapid and slow growth respectively), using an automated system (LiconiX incubator, robotic arm, and Tecan plate reader). We estimated *r* by fitting exponential equations to the first few data points representing the exponential growth phase from OD vs. time curves, using Curvefitter software (**Delaney et al., 2013**). After 8–12 hr of growth, we estimated the carrying capacity (K) by measuring the maximum OD of late log phase cultures (after 10 x dilution in rich media, to accurately estimate ODs higher than one). We estimated the length of the lag phase (L) as the time taken to reach the early log phase of growth. This was limited by low temporal resolution, and we were unable to capture differences in L that were smaller than 20 min (often observed in the WT and single gene deletions during rapid growth). We estimated the relative fitness of each mutant as the ratio of its r, K, or L value vs. that of the WT measured in the same experiment.

To measure the growth rate under nutrient shifts, we initiated precultures and sub-cultured them as above in a rich medium (TB). From late log phase culture in the rich medium (after 6 hr of growth), we again sub-cultured as above into poor medium (either M9 glycerol or M9 galactose, representing a nutrient downshift). When these cultures reached late log phase in the poor medium (12–16 hr), we again sub-cultured them back into the rich medium (TB, representing a nutrient upshift). After each transfer (downshift or upshift), we measured the growth rate as described above.

## Measuring mature tRNA pools using YAMAT-seq

For a subset of our tRNA gene deletion strains and WT, we measured the relative abundance of each tRNA isotype and compared this proportion across strains as described previously (**Ayan et al., 2020**; **Shigematsu et al., 2017**), or across growth media. We note that this procedure is analogous to measuring differences in relative mRNA levels by RNA-Seq. As such, it provides no information about (i) the absolute size of the mature tRNA pool, or (ii) the molecular mechanism(s) by which any observed differences arise (e.g. up versus down-regulation of gene expression, or degradation). Briefly, we grew three independent replicate cultures of each strain in two media and isolated total RNA from 4 ml (rich medium, LB) or 12 ml (poor medium, M9 +0.05% galactose) aliquots of mid-log phase cultures. We

extracted total RNA from each growing culture using a TRIzol Max Bacterial RNA isolation kit (Invitrogen; catalog number 16096040) as per the manufacturer's protocol. We subjected 10 µg of total RNA per sample to tRNA deacylation by incubating in 100 µl of 20 mM Tris-HCl (pH 9.0) for 40 min at 37 °C. We then desalted the deacylated RNAs and concentrated them by ethanol precipitation. Next, we took 1 µg of each precipitated product and ligated Y-shaped, DNA/RNA hybrid adapters (Eurofins; *Shigematsu et al., 2017*) to the conserved, exposed 5'-NCCA-3' and 3'-inorganic phosphate-5' ends of uncharged tRNAs using T4 RNA ligase 2 (dsRNA Ligase; New England BioLabs Inc, catalogue number M0239S). We reverse transcribed ligation products to cDNA using SuperScript III reverse transcriptase (Invitrogen; catalog number 18080093), and amplified cDNA products by eleven rounds of PCR with Phusion High Fidelity Master Mix with HF Buffer (New England BioLabs Inc, catalog number M0531S) with sample-specific indices (Illumina). We checked the quality and quantity of each PCR product using an Agilent DNA 7500 kit on a Bioanalyzer and a Florescence Nanodrop, respectively, and combined all samples in equimolar amounts into a single tube. We ran the mixture on a 5% native polyacrylamide gel, and excised bands between ~200 bp and ~280 bp size, which are expected to be enriched in tRNA-adapter sequences. Finally, we extracted the excised DNA in deionized water overnight, and removed agarose by centrifugation through filter paper. Quality and quantity of the final product were checked on a Fluorescence Nanodrop. The final product was sequenced by the sequencing facility within the Max Planck Institute for Evolutionary Biology, using an Illumina NextSeq 550 Output v2.5 kit (single-end, 150 bp reads). Raw reads and analysis files are available from NCBI GEO (accession number GSE198606).

We sorted raw reads for each sample using exact matches to each unique, 6 bp long Illumina barcode, obtaining a minimum of 707,429 reads per sample of which >99.99% were the expected length (80–151 bp) (*Supplementary file 3*). We assembled each set of reads to the 49 unique reference tRNA sequences predicted by GtRNAdb 2.0 (*Chan and Lowe, 2009*) for *E. coli* MG1655 (*Supplementary file 3*), allowing up to 10% mismatches, gaps of <3 bp, and up to five ambiguities per read. We discarded reads that aligned equally well to more than one tRNA sequence. Finally, we de novo aligned the unused reads for each sample and checked the resulting contigs to ensure that none contained substantial numbers of tRNA reads. We calculated the within-sample proportion of reads aligned to each tRNA type and mean mature tRNA isotype proportions for each strain across the three replicates. Finally, we used DESeq2 (*Love et al., 2014*) in R (version 3.6.0, *R Development Core Team, 2021*) to compare the fractional contribution of each tRNA isotype to the respective mature tRNA pool, across pairs of strains (e.g. WT vs. mutant) or media (rich vs. poor). We corrected for multiple testing using the Benjamini-Hochberg procedure (*Anders and Huber, 2010*). The raw YAMAT-seq reads and analysis files are available at the NCBI Gene Expression Omnibus (GEO accession number GSE198606) (*Edgar et al., 2002*).

## Measuring translation elongation rate

We measured the translation elongation rate for a subset of our strains, using the native β-galactosidase protein as a reporter as described earlier (*Miller, 1972*), with some modifications. Briefly, we induced *lacZ* gene expression in actively growing cultures ($OD_{600}$=0.5, n=2–3) with 0.5 mM isopropyl-β-D-thiogalactoside (IPTG). Every 15 s, we pipetted out 500 µl culture and immediately mixed it with 100 µl of chloramphenicol (3 mg/ml) to block translation. After 10 mins of incubation on ice, we added 350 µl of Z buffer (reaction buffer) and continued incubation on ice for 1 hr. Next, we added 200 µl of 12 mg/ml ONPG (o-nitro-phenyl galactopyranoside, substrate for β-galactosidase). After 1–1.5 hr of incubation at 30 °C to allow the full development of colored product (o-nitrophenol) due to enzyme activity, we stopped the reaction by adding 500 µl of 1 M $Na_2CO_3$. After a brief centrifugation step to collect debris (5000 g, 1 min), we transferred the supernatant to a 96-well microplate to assay the formation of o-nitrophenol by measuring $OD_{420}$. We converted OD values to Miller Units (MU) as per the original protocol, and from a plot of Miller Units (MU) of β-galactosidase vs. time, we estimated the first time point showing an increase in MU (after induction) as the time is taken to synthesize one molecule of β-galactosidase. The elongation rate (in amino acids per second) was inferred by dividing the length of the β-galactosidase protein (1019 amino acids) by this time.

The above assay informs us about the elongation rate, assuming that all other steps are equal between WT and mutant. Since, we only deleted genes encoding elongator tRNAs to create our mutant strains, this assumption likely holds. We also note that our WT strain contains an *rph* frameshift

mutation that leads to pyrimidine starvation (*Jensen, 1993*) in M9 medium. Therefore, we cannot make quantitative comparisons across media. However, comparing WT and mutants growing in the same medium is appropriate.

## Measuring translation capacity with a GFP reporter

We grew WT and mutants transformed with plasmid pKEN::GFP2 overnight in LB medium with ampicillin for 14–16 hr. We subcultured 1% v/v into M9 with 0.8% glucose and amino acids (with 100 μg/ml ampicillin) and incubated for around 4–5 hr (mid-log phase) before inducing with 0.5 mM IPTG. After induction, we measured GFP and $OD_{600}$ in a microplate reader (Infinite Pro Tecan, Austria). We first normalized GFP readout with OD at the same time point and then took the difference in GFP reading between time zero (immediately after induction) and after 1 hr to estimate the amount of GFP produced in an hour. We then normalized this readout for each mutant with the WT, to get the relative translation capacity of mutants.

## Data analysis

We used GraphPad Prism (9.0.0) for all statistical analysis (except for YAMAT-seq) and to generate heatmaps, and DEseq2 (*Love et al., 2014*) and R version 3.6.0 (*R Development Core Team, 2021*) for the analysis of YAMAT-seq data as described above. All raw data used for analysis are provided in source data files for each figure, and the statistical output is given in *Supplementary file 2*.

## Acknowledgements

We thank Joshua Miranda, Laasya Samhita, Shyamsunder Buddh, and Umesh Varshney for discussion and comments on the manuscript; Joshua Miranda, Laasya Samhita, Mrudula Sane, and the NCBS NGS facility for help with genome sequencing; Gaurav Diwan and Joshua Miranda for setting up and maintaining our automated growth measurement system; Gunda Dechow-Seligmann for helping with data collection for YAMAT-seq; and the NCBS laboratory kitchen staff for their crucial support, throughout this study and especially during the COVID-19 pandemic. We acknowledge funding and support from the National Centre for Biological Sciences (NCBS-TIFR) and the Department of Atomic Energy, Government of India (Project Identification No. RTI 4006) to DA, CSIR-UGC-NET June/2018/430 fellowship to PKR, the Max Planck Society (JG and WYN), and the International Max Planck Research School for Evolutionary Biology (WYN).

## Additional information

### Funding

| Funder | Grant reference number | Author |
| --- | --- | --- |
| Department of Atomic Energy, Government of India | Project 695 Identification No. RTI 4006 | Deepa Agashe Parth K Raval |
| CSIR-UGC | June/2018/430 | Parth K Raval |
| Max Planck Society | | Wing Yui Ngan |
| International Max Planck Research School for Evolutionary Biology | | Wing Yui Ngan |
| Max Planck Society | | Jenna Gallie |
| Deutsche Forschungsgemeinschaft | AOBJ:679030 | Wing Yui Ngan Jenna Gallie |

The funders had no role in study design, data collection and interpretation, or the decision to submit the work for publication.

### Author contributions

Parth K Raval, Conceptualization, Data curation, Formal analysis, Investigation, Visualization, Methodology, Writing - original draft, Writing – review and editing; Wing Yui Ngan, Data curation, Formal

analysis, Investigation, Visualization, Writing – review and editing; Jenna Gallie, Resources, Data curation, Formal analysis, Supervision, Funding acquisition, Visualization, Methodology, Writing – review and editing; Deepa Agashe, Conceptualization, Resources, Supervision, Funding acquisition, Visualization, Writing - original draft, Project administration, Writing – review and editing

**Author ORCIDs**
Parth K Raval ⓘ http://orcid.org/0000-0001-6151-437X
Jenna Gallie ⓘ http://orcid.org/0000-0003-2918-0925
Deepa Agashe ⓘ http://orcid.org/0000-0002-0374-8159

**Decision letter and Author response**
Decision letter https://doi.org/10.7554/eLife.81005.sa1
Author response https://doi.org/10.7554/eLife.81005.sa2

---

## Additional files

### Supplementary files
• Supplementary file 1. Detailed information on primers and strains used in this study, and the genomic location of tRNA genes.
• Supplementary file 2. Summary statistics for analyses associated with main figures.
• Supplementary file 3. Summary of YAMAT-seq data.
• MDAR checklist

### Data availability
All experimental data are provided as supplementary files. Raw YAMAT-seq reads and analysis files are available at the NCBI Gene Expression Omnibus (GEO accession number GSE198606).

The following dataset was generated:

| Author(s) | Year | Dataset title | Dataset URL | Database and Identifier |
|---|---|---|---|---|
| Gallie J, Ngan WY, Raval PK, Agashe D | 2022 | YAMAT-seq of mature tRNA pools in the bacterium *Escherichia coli* str. K-12 substr. MG1655 and derivatives | https://www.ncbi.nlm.nih.gov/geo/query/acc.cgi?acc=GSE198606 | NCBI Gene Expression Omnibus, GSE198606 |

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
