## [Editor Report]

The authors investigate the cost and benefits of maintaining seemingly redundant multiple copies of the translation machinery components. The authors demonstrate that while redundant multiple copies are beneficial in a nutrient-rich environment, maintaining these extra copies is costly and deleterious in a nutrient-poor environment. This explains why copy numbers of translation machinery genes are under selection according to the environmental niche an organism occupies. The work is very important and the findings exciting and supported by compelling evidence, in particular, the fitness gain upon deletion of translation genes in poor conditions is an insightful observation.

---

## [Decision Letter]

**Decision letter after peer review:**

Thank you for submitting your article "The layered costs and benefits of translational redundancy" for consideration by *eLife*. Your article has been reviewed by 3 peer reviewers, and the evaluation has been overseen by Yitzhak Pilpel as Reviewing Editor and Naama Barkai as the Senior Editor. The following individual involved in the review of your submission has agreed to reveal their identity: Dvir Schirman (Reviewer #3).

The reviewers have discussed their reviews with one another, and the Reviewing Editor has drafted this to help you prepare a revised submission. Each of the reviewers has comments for you to address (see below).

*Reviewer #1 (Recommendations for the authors):*

1) The induction lag method used to measure the "elongation rate" in Figure 5 quantifies the sum of the time to transcribe lacZ and the time to translate lacZ mRNA. To call it "elongation rate", the authors should stress the underlying assumption that all steps in b-galactosidase synthesis except elongation are equal in WT and mutant. Of note, MG16555 has an rph frameshift mutation that leads to pyrimidine starvation due to low pyrE expression levels (DOI: 10.1128/jb.175.11.3401-3407.1993). Thus, transcription is likely substantially faster in LB (contains pyrimidines) than in the M9 media.

2) The concentration of galactose used for poor quality growth medium is not mentioned in the methods.

3) Line 82: the reason rRNA concentration is more limiting for translation than tRNA is probably not that it is more abundant, but that rRNA concentration affects the concentration of ribozymes/catalysts for translation, while tRNA concentration affects the concentration of tertiary complex (the "substrate").

4) Line 317: "so that both rRNA and tRNA would be limiting" is a problematic statement. The levels of both would presumably be reduced, but not likely to both be limiting at the same time.

5) Line 336: presumably, the proM deletion mutant is growing OK because proL can read the same codon, not because the codon is rare.

6) Line 339: perhaps more likely than assigning different functionality to identical RNAs, the different deletions result in different concentrations of the tRNA due to e.g. polarity effects in transcription.

7) Line 464: what is meant by "reduced expression of the deleted tRNA"?

8) A compelling case that differences in tRNA levels and tRNA/rRNA ratios at different growth rates are largely determined by the location of the respective genes relative to the origin of replication was made recently for *E. coli* and other bacteria (Hu & Lercher, 2021, PLOS Genetics). Gene copy numbers of genes close to the origin are more sensitive to growth rate than genes located close to the terminus. This work would benefit from including the distance to origin in the analysis of the impact of various gene deletions.

Regarding tRNA measurements by YAMAT-seq:

The authors should express more clearly in which sense the measurements are "relative", as it is not the relative expression level but the relative fractional contribution of each tRNA to the respective tRNA pool that is reported (e.g. line 254 is incorrect : DEseq2 was used to measure tRNA expression level differences between pairs of strains"; line 922 is incorrect: "Expression level of all 42 WT tRNA isotypes in poor medium relative to rich medium", and others, including much of the wording in section "Gene regulation cannot compensate for the loss of tRNA gene copies").

A final note is that the reason YAMAT-seq does not report on levels of tRNAs is because a fixed amount of total RNA was used for the procedure (10 ug from each sample). Fast-growing cells have more RNA than slow-growing cells (both more RNA per cell, more RNA per OD, more RNA per ug protein, etc), so 10 ug represents RNA from more cells in the slow-growing cultures than it does in the fast-growing cultures.

Line 42: it is not completely clear to this reviewer what is meant by "target" and "non-target" tRNAs. Further explanation in the Introduction would be beneficial for some readers (like myself).

Line 282: was a pBAD plasmid really induced with IPTG, or is it a typo? I am only familiar with the arabinose-induction of pBAD plasmids.

Figure 2 - Figure Supplement 1 1: Growth curves on a linear scale cannot be used to evaluate growth rates by eye. Showing the curves on semi-log graphs would be much more informative to the reader.

*Reviewer #2 (Recommendations for the authors):*

1. Please be clear and consistent in referencing the figure.

2. Please address or better explain your methodology, especially how Rrel can be negative.

3. Please change figure 3 to mark significant and non-significant data points and discuss them in the text.

4. Please tone down your arguments in places where they sound general when, in effect, they are specific to a condition (e.g., figure 3).

5. I am missing controls. Please express deleted tRNAs from a plasmid in at least some deletion strains. If the author's hypothesis is true, it will rescue the negative fitness observed in deletion strains in rich media. These controls are needed at least in Figures 2 and 3. Notably, I would like to stress again that it should be done on a subset of the strains, probably the ones that display the strongest effect.

*Reviewer #3 (Recommendations for the authors):*

1. I think that in the discussion part the authors can discuss the possible cost/benefit effects of deleting one of the two copies of the translation elongation factor EF-Tu (Zuurmond et al. 1999, doi: 10.1007/s004380050934). As this is another example of seeming redundancy of the translation machinery in *E. coli*, and since EF-Tu is known as the most abundant cytoplasmic protein in *E. coli* I think it is an interesting case of translation machinery redundancy, and addressing it should benefit the manuscript.

2. In the discussion the authors suggest a model that explains their results (line 636). I think this suggested model will be clearer and easier to understand if it is described also through a graphical illustration.

3. In Figure 2D, I think the figure will be easier to understand with a clearer example of the relevant correlation using real data points. I would replace the bottom right example with one of the panels from Figure 2 —figure supplement 5.

4. In Figure 3 both panels have the same axes labels (Glycerol to TB / TB to glycerol), according to the figure legend one of them should be switched to (Galactose to TB / TB to galactose).

---

## [Author Response]

Reviewer #1 (Recommendations for the authors):1) The induction lag method used to measure the "elongation rate" in Figure 5 quantifies the sum of the time to transcribe lacZ and the time to translate lacZ mRNA. To call it "elongation rate", the authors should stress the underlying assumption that all steps in b-galactosidase synthesis except elongation are equal in WT and mutant. Of note, MG16555 has an rph frameshift mutation that leads to pyrimidine starvation due to low pyrE expression levels (DOI: 10.1128/jb.175.11.3401-3407.1993). Thus, transcription is likely substantially faster in LB (contains pyrimidines) than in the M9 media.

Thank you for pointing this out. We have now mentioned both points in the methods section “Measuring translation elongation”

2) The concentration of galactose used for poor quality growth medium is not mentioned in the methods.

We have now added this in the methods section "Measuring growth parameters”.

3) Line 82: the reason rRNA concentration is more limiting for translation than tRNA is probably not that it is more abundant, but that rRNA concentration affects the concentration of ribozymes/catalysts for translation, while tRNA concentration affects the concentration of tertiary complex (the "substrate").

Agreed, we have now incorporated this (line 83-84).

4) Line 317: "so that both rRNA and tRNA would be limiting" is a problematic statement. The levels of both would presumably be reduced, but not likely to both be limiting at the same time.

We have modified the sentence to just say that they will both be reduced (line 338-39).

5) Line 336: presumably, the proM deletion mutant is growing OK because proL can read the same codon, not because the codon is rare.

The deletion mutant mentioned here is proL (not proM). The proL tRNA is a single-copy tRNA with no tRNA (or ME) backup. Hence, we speculate that its loss has negligible effects because the cognate codon is only used rarely.

6) Line 339: perhaps more likely than assigning different functionality to identical RNAs, the different deletions result in different concentrations of the tRNA due to e.g. polarity effects in transcription.

By functional differences we meant that at the level of contribution to translation the two tRNAs are different, e.g. due to polarity effects as mentioned in the comment. We did not mean to say the tRNAs have completely distinct functions (outside the context of translation). We have now edited the sentence to avoid confusion (lines 361-362).

7) Line 464: what is meant by "reduced expression of the deleted tRNA"?

Here, we refer to the deletion of single-copy tRNA genes, which should lead to a complete loss of the relevant tRNA isotypes. In YAMAT-seq data, this will show up as a large drop in the relative contribution of these tRNA isotypes to the tRNA pool (compared to WT). We have now rephrased the text here to avoid confusion (lines 488-489). We have also ensured that throughout the manuscript we have clarified whether we are referring to tRNA genes or tRNA isotypes.

8) A compelling case that differences in tRNA levels and tRNA/rRNA ratios at different growth rates are largely determined by the location of the respective genes relative to the origin of replication was made recently for *E. coli* and other bacteria (Hu & Lercher, 2021, PLOS Genetics). Gene copy numbers of genes close to the origin are more sensitive to growth rate than genes located close to the terminus. This work would benefit from including the distance to origin in the analysis of the impact of various gene deletions.

Thank you for this suggestion. The paper mentioned above also led us to other useful references, which we now cite in the manuscript.

Distance from the replication origin does not explain differences in the fitness impacts of copies of the same tRNA isotype (e.g. Asn-GUU). We have now included this in the text (lines 362-367) and in a new supplemental figure (Figure 2 supplement 3).

Regarding tRNA measurements by YAMAT-seq:The authors should express more clearly in which sense the measurements are "relative", as it is not the relative expression level but the relative fractional contribution of each tRNA to the respective tRNA pool that is reported (e.g. line 254 is incorrect : DEseq2 was used to measure tRNA expression level differences between pairs of strains"; line 922 is incorrect: "Expression level of all 42 WT tRNA isotypes in poor medium relative to rich medium", and others, including much of the wording in section "Gene regulation cannot compensate for the loss of tRNA gene copies").A final note is that the reason YAMAT-seq does not report on levels of tRNAs is because a fixed amount of total RNA was used for the procedure (10 ug from each sample). Fast-growing cells have more RNA than slow-growing cells (both more RNA per cell, more RNA per OD, more RNA per ug protein, etc), so 10 ug represents RNA from more cells in the slow-growing cultures than it does in the fast-growing cultures.

Thank you very much for these thoughtful comments. We agree that the phrasing was incorrect and confusing. We have now clarified in the methods section that YAMAT-seq is like RNA-seq, and only measures the contribution of a given tRNA isotype to the total tRNA pool (lines 221-225). We have also rephrased the text in the entire results section about YAMAT-seq as well as figure legends, to clearly convey the comparisons being made.

Line 42: it is not completely clear to this reviewer what is meant by "target" and "non-target" tRNAs. Further explanation in the Introduction would be beneficial for some readers (like myself).

This was described with an example in the Introduction (lines 45-52). We have now edited the text there to explain why we call them “target” tRNAs, which is hopefully helpful; and we have now added more explanation in the legend for figure 1.

Line 282: was a pBAD plasmid really induced with IPTG, or is it a typo? I am only familiar with the arabinose-induction of pBAD plasmids.

This was a mistake, thank you for pointing it out. It was a pKEN plasmid, not pBAD; we have now corrected this.

Figure 2 - Figure Supplement 1 1: Growth curves on a linear scale cannot be used to evaluate growth rates by eye. Showing the curves on semi-log graphs would be much more informative to the reader.

We have now added growth curves on a log scale to this supplementary figure.

Reviewer #2 (Recommendations for the authors):1. Please be clear and consistent in referencing the figure.2. Please address or better explain your methodology, especially how Rrel can be negative.3. Please change figure 3 to mark significant and non-significant data points and discuss them in the text.4. Please tone down your arguments in places where they sound general when, in effect, they are specific to a condition (e.g., figure 3).5. I am missing controls. Please express deleted tRNAs from a plasmid in at least some deletion strains. If the author's hypothesis is true, it will rescue the negative fitness observed in deletion strains in rich media. These controls are needed at least in Figures 2 and 3. Notably, I would like to stress again that it should be done on a subset of the strains, probably the ones that display the strongest effect.

Thank you for these points, which we have addressed as detailed above.

Reviewer #3 (Recommendations for the authors):1. I think that in the discussion part the authors can discuss the possible cost/benefit effects of deleting one of the two copies of the translation elongation factor EF-Tu (Zuurmond et al. 1999, doi: 10.1007/s004380050934). As this is another example of seeming redundancy of the translation machinery in *E. coli*, and since EF-Tu is known as the most abundant cytoplasmic protein in E. coli I think it is an interesting case of translation machinery redundancy, and addressing it should benefit the manuscript.

Thank you for this suggestion, we have now discussed this work (lines 693-696).

2. In the discussion the authors suggest a model that explains their results (line 636). I think this suggested model will be clearer and easier to understand if it is described also through a graphical illustration.

Thank you for this suggestion. We have now added a graphical representation as (new) Figure 7.

3. In Figure 2D, I think the figure will be easier to understand with a clearer example of the relevant correlation using real data points. I would replace the bottom right example with one of the panels from Figure 2 —figure supplement 5.

Here, we wanted to show the possible relationships and various implications, which would be difficult if we used actual data. Also, given the large variation across panels in the supplemental figure, we worry that choosing one would be misleading. Hence, we have retained the schematic.

4. In Figure 3 both panels have the same axes labels (Glycerol to TB / TB to glycerol), according to the figure legend one of them should be switched to (Galactose to TB / TB to galactose).

Corrected, thank you.